# Prospects of Microalgae for Biomaterial Production and Environmental Applications at Biorefineries

**Lourdes Orejuela-Escobar** [1,2,3,*], **Arleth Gualle** [1] , **Valeria Ochoa-Herrera** [2,4] **and George P. Philippidis** [5,*]

1   GICAS Laboratory, Department of Chemical Engineering, College of Sciences and Engineering, Universidad San Francisco de Quito, Quito 170901, Ecuador; arleth.gualle@alumni.usfq.edu.ec
2   Instituto Biosfera, College of Sciences and Engineering, Universidad San Francisco de Quito, Quito 170901, Ecuador; vochoa@usfq.edu.ec
3   Instituto de Investigaciones Biomedicas, Universidad San Francisco de Quito, Quito 170901, Ecuador
4   Department of Environmental Sciences and Engineering, Gillings School of Global Public Health, University of North Carolina, Chapel Hill, NC 27599, USA
5   Patel College of Global Sustainability, University of South Florida, 4202 E. Fowler Avenue, Tampa, FL 33620, USA
*   Correspondence: lorejuela@usfq.edu.ec (L.O.-E.); gphilippidis@usf.edu (G.P.P.)

**Abstract:** Microalgae are increasingly viewed as renewable biological resources for a wide range of chemical compounds that can be used as or transformed into biomaterials through biorefining to foster the bioeconomy of the future. Besides the well-established biofuel potential of microalgae, key microalgal bioactive compounds, such as lipids, proteins, polysaccharides, pigments, vitamins, and polyphenols, possess a wide range of biomedical and nutritional attributes. Hence, microalgae can find value-added applications in the nutraceutical, pharmaceutical, cosmetics, personal care, animal food, and agricultural industries. Microalgal biomass can be processed into biomaterials for use in dyes, paints, bioplastics, biopolymers, and nanoparticles, or as hydrochar and biochar in solid fuel cells and soil amendments. Equally important is the use of microalgae in environmental applications, where they can serve in heavy metal bioremediation, wastewater treatment, and carbon sequestration thanks to their nutrient uptake and adsorptive properties. The present article provides a comprehensive review of microalgae specifically focused on biomaterial production and environmental applications in an effort to assess their current status and spur further deployment into the commercial arena.

**Keywords:** microalgae; biomaterials; bioeconomy; biorefinery; environmental applications

## 1. Introduction

Microalgae are a diverse group of organisms that thrive in various habitats, including rivers, lakes, oceans, and soils. Thanks to their photosynthetic efficiency, they enjoy high growth rates and can be potentially cultivated in sustainable ways [1]. They have certain requirements for growth, such as light, nutrients, and appropriate pH, and are capable of obtaining energy from photosynthesis and organic carbon, when growing under photoautotrophic and heterotrophic or mixotrophic conditions, respectively [2]. In recent years, microalgae have been investigated as a potential source of biomaterials useful for the energy, nutraceutical, pharmaceutical, cosmetic, agriculture, bioremediation, and agriculture industries. Algal cells contain numerous metabolites and bioactive com-pounds, such as polysaccharides, sugars, proteins, lipids, carotenoids, and pigments, which have functional properties that make them promising agents for biotechnological and environmental applications [3]. Specific applications include production of biofuels (ethanol, butanol, biogas, hydrogen), food (animal feed, nutraceuticals, hydrocolloids), pharmaceuticals (antiviral, antifungal, neuroprotective, skin antiaging agents, UV protectants), fertilizers and soil amendment, and chemical feedstocks (pigments, dyes, colorants, biopolymers,

bioplastics, nanoparticles) and use in environmental remediation (carbon capture, removal of nutrients) [4,5].

The production of microalgal biomass has advantages over traditional crops: (1) high productivity, approximately 10–100× that of terrestrial crops and (2) high content of lipids and other bioactive components of commercial interest. Moreover, microalgae can be cultivated sustainably utilizing wastewater, brackish water, or seawater in batch, fed-batch, and continuous production mode on non-arable land, hence reducing potable water use, land use change, and competition with food [6]. Following cultivation, cell harvesting is crucial in terms of cost and energy intensity. It can be performed via a variety of physico-chemical techniques, such as filtration, flotation, centrifugation, flocculation, and coagulation, depending on the algal species and the algal product of interest [7]. Although natural settling is an option, it is impractical as it takes place slowly, but can be expedited by applying bioflocculation with the use of air bubbling or shaking [8]. An integrated approach involving high-productivity cultivation and efficient harvesting can help advance the status of algae technologies [9]. Extraction of biochemical components of interest from microalgae requires cell disruption and fractionation methods. The cell wall is rigidly built with glycoproteins, polysaccharides, such as pectin, cellulose, and hemicellulose, and other components, like silica, lignin, and carbonate, hence requiring physical and chemical processing to break down, such as shear, cavitation, pulsed electric field, hydrolysis, enzymatic digestion, extraction with supercritical fluids, use of eutectic solvents, and high pressure homogenization [4].

Microalgae-based technologies still face a number of hurdles, such as cost competitiveness and sustainability, before reaching commercial scale [10]. The cultivation and downstream processing of microalgae present challenges for the algae biorefineries of the future, such as reducing the media cost, diminishing contamination risks, developing more efficient techniques for fast and high-yield extraction of active compounds, and achieving high selectivity for the products of interest. Intensive research and development in the algae arena has been taking place around the world to improve productivity and reduce operating costs, as demand for value-added bioproducts is constantly growing and consumer acceptance is strengthening [11]. The present review focuses on the promising role that algae biorefineries can play in the sustainable bioeconomy of the future and outlines the cellular functions, health benefits, and industrial applications of microalgae as a renewable feedstock for the production of biomaterials, and as a remediation agent in environmental applications.

## 2. Microalgae as a Renewable Industrial Feedstock

### 2.1. Biorefinery as the Basis of the Bioeconomy

The UN-adapted Sustainable Development Goals (SDG) include the development of a bioeconomy whose building blocks for materials, chemicals, and energy are derived from renewable biological resources rather than from fossil resources, like oil and coal [12]. Furthermore, a circular bioeconomy seeks to integrate production processes making use of residues and wastes to reduce virgin material, energy, and water consumption via cascading [13]. In essence, the bioeconomy is the response of social awareness to raw material scarcity and ecosystem deterioration by promoting environmental stewardship and sustainable manufacturing.

The cornerstone of the bioeconomy is the biorefinery, which is analogous to an oil refinery. It refers to a chemical facility that carries out a series of integrated processes with the purpose of profitably and sustainably fractionating renewable algal or terrestrial biomass into a plethora of intermediate and final products, primarily biofuels and bioproducts, for use in the economy. A biorefinery entails a number of raw materials, intermediate products, and final products that are processed or produced by several unit operations [1,14]. At the same time, biorefinery processes need to operate sustainably by minimizing water and energy use and waste generation [15,16]. Zero waste efforts treat residues to secondary

renewable feedstocks for other processes through reduction, recycling, reuse or recovery of solid and liquid byproducts resulting from microalgal biomass production [15].

Biorefineries have evolved significantly since the 1980s, as depicted in Figure 1. The first generation biorefinery is based on food carbohydrates, mainly sugar cane and corn in the Americas [17], and cassava in other parts of the world [18]. The second generation biorefinery is more sustainable as it shifts away from food towards inedible renewable biomass sources for obtaining biofuels and bioenergy, such as woody biomass, agricultural residues, and grasses, as well as municipal and urban waste [19]. It should be noted that municipal waste that ends in landfills varies considerably in composition depending upon the level of urbanization and industrialization of the area of origin [20]. In the last decade, the third generation biorefinery has evolved using fast-growing photosynthetic microalgae as raw material for biofuels and bioproducts, such as pigments, lipids, proteins, and carbohydrates [21]. Finally, the fourth generation biorefinery incorporates genetically modified algae and cellulosic biomass to further enhance productivity through overexpression of metabolites and enzymes, and thus improve the economics [22,23].

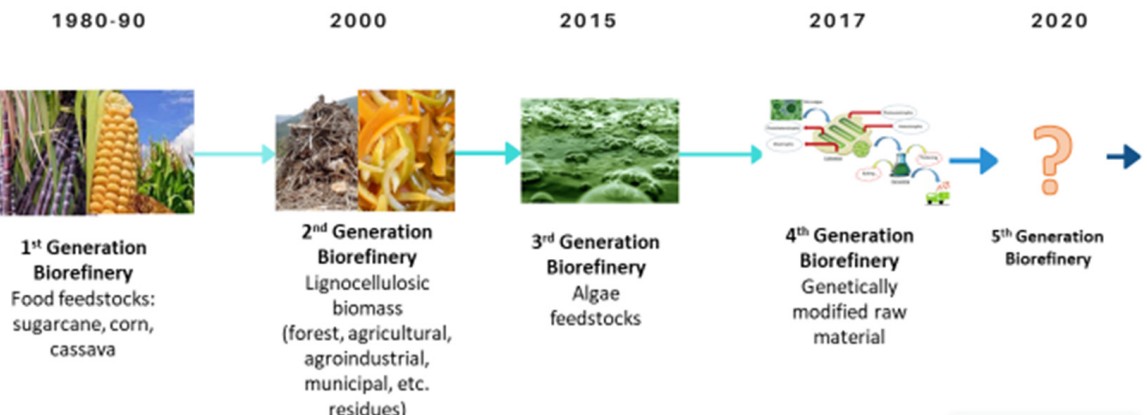

**Figure 1.** Evolution of biorefinery feedstocks.

The objective of third-generation algal biorefineries is to sustainably cultivate microalgae and process them into a broad spectrum of bioproducts marketed as advanced biofuels and biomaterials that will replace their fossil counterparts, thus helping reduce carbon emissions, and combat climate change and environmental pollution. The success of the biorefinery depends on the availability of proven and scalable processes necessary to fractionate and purify biological components derived from algal biomass, such as lipids, carbohydrates, proteins, and bioactive compounds, as detailed in Figure 2. Moreover, such processes have to be cost-effective to lead to profitable operation of the biorefinery and sustainable to contribute to the bioeconomy's goals of sustainability and resiliency with regards to the triple bottom line of environment, economy, and society.

Microalgae biorefineries can serve the energy, nutraceutical, cosmetics, medical, and food industries by using appropriate microalgae strains and cultivation conditions to produce the desired products [21]. As outlined in Figure 3, the key process steps and parameters of a biorefinery are: (1) Cultivation: type of strain, operating conditions, supply of carbon dioxide, source of nutrients, and source of lighting to optimize growth and product formation; (2) Biomass harvesting: dewatering method and conditions; and (3) Biomass processing: cell disruption through treatment, metabolite recovery, and conversion to biomaterials [24].

While biofuels undoubtedly represent the largest market opportunity for microalgae, the volatility of fossil fuel prices presents a major challenge to biofuel viability that can be addressed in the long term only if biorefineries complement biofuels with the co-production of value-added biomaterials to strengthen and diversify their cash flow [10,25,26]. As shown in Figure 2, biomaterials that can be manufactured in an algal biorefinery include various bioactive compounds, such as polyphenols, carotenoids, and vitamins, for use as

nutraceuticals, cosmetics, pharmaceuticals, functional foods, and agrochemicals. Lipids and polyunsaturated fatty acids (PUFA) are used for functional food applications, while polysaccharides are the backbone of a wide range of products including bioplastics, and proteins serve as animal feed, fish meal, and soil amendment.

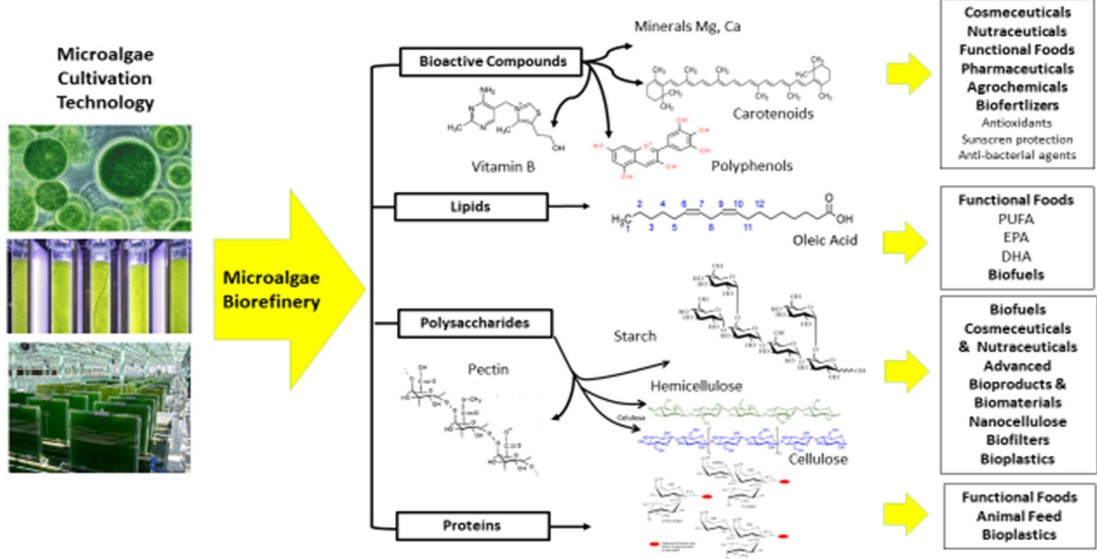

**Figure 2.** Product portfolio of third generation biorefinery.

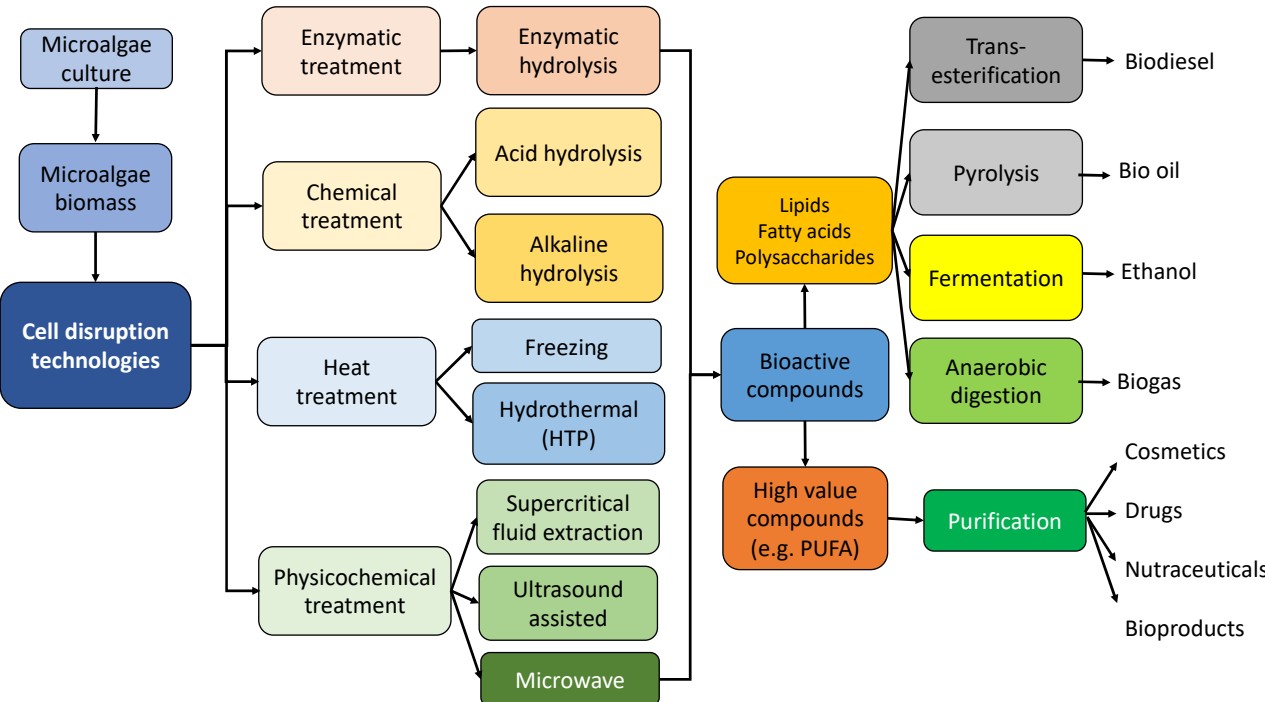

**Figure 3.** Key biorefinery processes for algal cell disruption, processing, and product recovery.

### 2.2. Microalgae Processing Technologies

Algae cultivation, conducted in open raceway ponds or photobioreactors, has been extensively covered in the literature [15,27,28] and will not be repeated here. Obtaining the final microalgal product cost-effectively and sustainably in a biorefinery requires a series of sequential unit operations depending on the particular species of microalgae employed [25].

As seen in Figure 3, a variety of technologies are applied to algae biomass to rupture the cell wall and recover the products of interest from the cellular content. Physicochemical treatments use microwaves, ultrasound, or supercritical fluid extraction. The microwave frequency more commonly used is 2450 MHz and causes intracellular water to boil and increase pressure, thus rupturing the cells. Ultrasound technology uses shock waves to generate cavitation bubbles that break the cell wall. In extraction with supercritical fluids, $CO_2$ is generally used at a temperature and pressure higher than the critical point to destroy the cell wall and release metabolites.

Chemical treatments use various chemicals, such as chelating agents, surfactants, solvents, hypochlorites, acids, and alkalis, aiming at disrupting the microalgal cell wall and membrane. For example, solvents dissolve cell wall components, alkalis saponify membrane lipids, whereas acids create pores, allowing intracellular substances to be released. Enzymatic hydrolysis, on the other hand, is a biological treatment of higher specificity that does not require high use of energy. Enzymes like glucosidases, glucanases, peptidases, and lipases can degrade the cell wall [1].

### 3. Cellular Components of Interest and Their Role in Microalgal Metabolism

#### 3.1. Proteins

Microalgae are a rich source of protein similar to food sources, such as meat, eggs, or soy. The quality of algal protein depends on the content of essential amino acids that serve various functions [29]. Higher cellular protein content is associated with an increase in light irradiation over a long photoperiod, whereas the efficiency of protein extraction is influenced by the structural characteristics, chemical composition, and morphology of the microalgae [30]. Several species of microalgae have been identified as high-protein producers, including phycobiliproteins and glycoproteins: *Spirulina* sp. contain up to 70% protein by dry weight with a high percentage of digestibility, *Chlorella vulgaris* contains 55%, and *Dunaliella* sp. 57% [31]. Recently, a 25% protein content was reported for the freshwater microalga *Scenedesmus obliquus* BR003 [32].

#### 3.2. Enzymes

Numerous enzymes, which are proteinaceous molecules that catalyze metabolic reactions, are synthesized in microalgae [30]. They include cellulases, lipases, amylases, galactosidases, proteases, phytases, and laccases. The presence of cellulases, hemicellulases, and pectinases provides capacity to degrade polysaccharides. Amylases, such as β-amylase, glucoamylase, isoamylase, and glucosidases, promote the hydrolysis of starch. The enzyme galactosidase hydrolyzes galactose residues linked to the α-1,6 bond found in oligosaccharides and galactomannans [33]. Microalgal laccases, which promote complex polymer oxidation, degradation, and lignin biosynthesis, are involved in the transformation of various natural aromatic and xenobiotic compounds [34]. Microalgae synthesize enzymes needed for synthesis of fatty acids and modification through desaturation and elongation, such as acyl-CoA, diacylglycerol, acyltransferase 2, and Δ6-desaturase, which can be obtained from a variety of species, such as *Chlamydomonas reinhardtii*, *Chlorella elipsoidea*, and *Phaeodactylum tricornutum* [30]. Neutral lipids are synthesized by esterification of glycerol with fatty acids catalyzed by lipases, and such lipids, once extracted from microalgae, can be converted to biodiesel via alkali-catalyzed transesterification in the presence of methanol or ethanol. Moreover, microalgal enzymes are used in bioremediation as is the case with chromate reductases that convert chromium into a less toxic form, while other enzymes can bioremediate cyanide. *Chlamidomonas* sp. and *Chlorella vulgaris* are sources of the enzyme L-asparaginase used for treatment of acute lymphoblastic leukemia, acute myeloid leukemia, and non-Hodgkin lymphoma [30].

#### 3.3. Sugars and Polysaccharides

The algal cell wall contains sugars and polysaccharides that help form a rigid barrier in conjunction with membrane lipids [35]. Various factors affect the amount of carbohydrates

present in the cell wall, such as pH, light, salinity, and availability of nutrients. Some carbohydrates are also present in vacuoles or are secreted as exopolysaccharides. Microalgae generally contain more xylose, glucose, and galactose than mannose, methylated galactose, and other pentoses [31]. The composition and proportion of polysaccharides also vary with morphology, growth phase, and life stage of the organism. They fulfill various types of functions within microalgal cells, mainly as a source of energy and protective functions. Among the main types of polysaccharides that have been identified in microalgae are starch, glycogen, cellulose, alginate, and agar [36]. There are numerous medical and industrial applications of polysaccharides from microalgae. Calcium spirulan shows antiviral activity by inhibiting the replication of viruses and preventing the proliferation of tumor cells, whereas glycosoaminoglycan obtained from *Porphyridium* sp. is an anti-inflammatory, and fucose and rhamnose find applications in the food industry as thickening agents, food additives, and stabilizers [37].

### 3.4. Fatty Acids and Lipids

Algal lipids are located in intracellular droplets that contain mainly triacylglycerides (TAG) and are encapsulated by a membrane consisting of phospholipids, glycolipids, and proteins. TAG, which are used in the production of biofuels, help algal cells manage oxidative stress or other adverse environmental conditions, and usually represent 10–40% of dry cell weight [38]. Microalgal lipids are classified into polar phospholipids and glycolipids and neutral triglycerides, fatty acids, and sterols [11].

These molecules play a fundamental role in the metabolism and growth of microalgae given that phosphoglycerides, glycosylglycerides, and sterols are essential structural components of the cell membrane, polyunsaturated fatty acids act as intermediaries for cell signaling pathways and play a role in detecting environmental changes, and neutral lipids in general are a means of storing energy necessary to carry out various cellular activities [39]. Lipids are mainly composed of glycerol and unsaturated fatty acids with chain lengths of 14 to 20 carbon atoms, but some species can produce fatty acids with carbon chains greater than 20 that are mostly polyunsaturated fatty acids (PUFA) [40]. The composition of lipids is variable and depends on growth conditions, nutrient availability, temperature, light intensity, pH, aeration rate, and light–dark cycle ratio [41].

### 3.5. Carotenoids and Pigments

Carotenoids are soluble lipid pigments that have antioxidant properties offering algae cells protection from the effect of free radicals, when they are exposed to high oxygen and light levels [25]. In addition, carotenoids also mediate collection and transmission of light to chlorophyll molecules. Among the main carotenoids synthesized by microalgae, the red pigment astaxanthin is produced in significant amounts by *Haematococcus pluvialis* at concentrations of 1.5–4% of dry cell weight [3]. *Dunaliella salina* is known for significant synthesis of β-carotene and *Scenedesmus* sp. produce large amounts of lutein, an antioxidant carotenoid [42].

The synthesis of carotenoids is a complex process involving a series of genes and enzymes [43]. An effective way to induce their expression in algae is through various types of stress, such as light intensity and nutrient limitation. Light irradiation causes formation of reactive oxygen species (ROS) that trigger carotenoid synthesis for cell protection. Similarly, nutrient limitations lead to production of carotenoids as a defense mechanism [44].

### 3.6. Polyphenols

Polyphenols are also antioxidants found in microalgal extracts at microgram levels comparable to those found in terrestrial plants [45]. Studies in *Spirulina maxima* demonstrated the effect of nutrients in the culture medium on the synthesis of phenolic compounds, such as salicylic, transcinnamic, chlorogenic, and caffeic acids [46]. Other species synthesizing significant amounts of polyphenols are *Phaeodactylum tricornutum* and *Tetraselmis suecica* [47].

### 3.7. Vitamins

Microalgae synthesize a wide variety of water-soluble and lipid-soluble vitamins, including A, D, E, K, C, B1, B2, B3, B5, B6, B7, B9, and B12, with vitamins A, B3, E, and B9 (folic acid) being produced in higher amounts [48]. Vitamins are essential to algal metabolism as enzyme cofactors, hormones, antioxidants, mediators of cell signaling, and regulators of tissue growth or differentiation [48]. The vitamin content of microalgae varies with environmental factors, nutrient concentration, harvest and drying method, and type of extraction. Studies indicate that microalgae secrete amounts of vitamins that are comparable to those in certain fruits and vegetables, therefore creating interest in extraction of such bioactive compounds in a biorefinery [49].

The species with the highest reported vitamin content are *Chlorella* and *Spirulina*, which produce beta-carotene, vitamin C, vitamin E, thiamine, riboflavin, niacin, pantothenic acid, pyridoxine, folic acid, and cobalamin. Particularly high levels of folic acid and thiamine were found in *Chlamydomonas* and *Chlorococcum* sp. [50]. High concentrations of folates were also found in *Chlorella vulgaris* with a concentration of 3.5 mg/100 g of dry cell mass, whereas the highest yield was obtained from marine *Picochlorum* sp. at 6.5 mg/100 g [51].

## 4. Health Properties of Microalgal Metabolites

### 4.1. Antioxidant Activity

The antioxidant capacity of microalgae is equal to or greater than that of plants or fruits as they contain a wide range of molecules, such as polyphenols, carotenoids, pigments, sterols, and vitamins that scavenge for free radicals and ROS [52]. Several microalgae have demonstrated antioxidant potential, when tested on 1,1-diphenyl-2-picrylhydrazyl (DPPH), superoxide anion, hydroxyl radical, and peroxyl radical [31]. Some studies have revealed the presence of antioxidant effects in aqueous and methanolic extracts of various microalgae species [45].

Among phenolic compounds with antioxidant capacity flavonoids are an important group, whereas sterols show effective results against neurological diseases. In addition, microalgal vitamins E, B3, B1, biotin, and riboflavin are beneficial to human health as they help secure normal functioning of cells and tissues, including the presence of polysaccharides, sulfates, enzymes, peptides, and amino acids [53]. The carotenoid astaxanthin is also a popular antioxidant used in food, pharmaceutical, and cosmetic products with an activity that is $10\times$ greater than β-carotene and $1000\times$ greater than vitamin E. Among its health benefits are anti-aging and suppression of age-related diseases, boosting of the immune system, and support of heart, liver, and joint health [54].

### 4.2. Antitumor Activity

Various microalgal bioactive compounds show antitumor activity by modulating various human cellular mechanisms, such as cytotoxicity, tumor cell propagation, and proliferation and apoptosis of cancer cells. Such compounds include carotenoids, PUFA, polysaccharides, and peptides [55]. The antioxidant carotenoids have also been correlated with the treatment of cancer with astaxanthin, showing potential for preventing the proliferation of gastric cancer cell lines in humans and fucoxanthin disrupting cancer cell growth and stimulating suppressor genes without affecting tumor cell apoptosis [56].

Polysaccharides extracted from microalgae can also serve in cancer treatment, with their biological activity depending on molecular weight, monosaccharide composition, and even sulfate content, as sulfated polysaccharides may be more effective for use in medical treatments [57]. Fucoidan, a sulfated polysaccharide enriched with fucose and isolated from brown algae, has shown an antitumor effect and an antiproliferative impact on Lewis lung cancer in mice that is correlated to sulfate content [58].

### 4.3. Antimicrobial Activity

Some microalgal metabolites demonstrate antimicrobial activity by targeting peptides, sterols, alkaloids, and amides as a defense mechanism against bacteria, fungi, and viruses,

when they are present in their environment [59,60]. One of the first antimicrobial substances isolated from microalgae was chlorellin, a mixture of fatty acids extracted from *Chlorella* sp. This mixture inhibited the development of Gram positive and Gram negative bacteria by slowing down the growth of *Staphylococcus aureus* and *Bacillus cereus* strains [61]. Such bioactive components of microalgae can be potentially beneficial in medical treatments as replacement of traditional synthetic drugs.

### 4.4. Antifungal Activity

There is ongoing research on microalgae strains and compounds that exhibit antifungal activity on various food and medical pathogens, as seen in crude extracts of microalgae generated using various solvents and types of extraction [62]. *Spirulina maxima* extract showed activity against various human and plant pathogenic fungi [63]. Antifungal activity is believed to be due to butanoic acid and methyl lactate, which inhibit fungal growth [64]. *Chlorella pyrenoidosa* can be used for food preservation to control fungal proliferation [65]. Culture supernatants of *C. vulgaris* and *C. ellipsoidea* showed antifungal activity against *Candida* sp. and *Aspergillus* sp., whereas extracts of *D. salina* had inhibitory activity against *A. nigers* and *C. albicans* [66]. Extracts of *Chlorella vulgaris* and *Chlorella minutissima* showed strong antifungal activity against *Aspergillus niger* and *Fusarium oxysporum* [67]. Recently, antifungal activity was reported in extracts of *Nannochloropsis oculata* against *Candida albicans*, where the bioactive compounds were terpenoids, carotenoids, polyphenols, and fatty acids [68].

### 4.5. Anti-Inflammatory Activity

As traditional anti-inflammatory drugs generally show undesirable adverse effects in patients, there has been interest in identifying natural alternatives. Anti-inflammatory properties have been identified in antioxidant microalgal components, including carotenoids, carbohydrates, and polyunsaturated fatty acids [69]. They show promise in cases of arthritis, gastrointestinal disorders, and atherosclerosis. Lycopene isolated from a *Chlorella* strain showed anti-inflammatory effects against arthritis in studies carried out in rats [31]. Marine microalgae seem to possess potential against neuroinflammatory diseases by acting at various cellular levels, inhibiting pro-inflammatory enzymes, modulating pathways, and controlling immune responses, as shown in vitro studies conducted on animals [70]. Phycobiliproteins, phenolic compounds, and various isomers of carotenoids have also been investigated in anti-inflammatory applications [71].

### 4.6. Antiviral Activity

Marine microalgae have been found to exhibit antiviral properties. Sulfated extracellular polysaccharides of *Cochlodinium polykrikoides* inhibited the influenza virus, the human immunodeficiency virus type 1, and the herpes simplex virus in mammalian cells [72]. Polysaccharides from *Porphyridium* sp. exhibited antiviral effects on *Herpes simplex* and *Varicella zoster* [73]. Additionally, they also inhibited malignant cell transformation, when added before or during the infection, suggesting that the inhibitory effect of the polysaccharides was due to the blocking effect on viral receptors, preventing viral penetration into mammalian cells, and interfering with retrovirus replication [74].

Sulfated polysaccharides are negatively charged due to the sulfate groups and have monosaccharide groups distributed along the polysaccharide chain, which may influence the specificity of protein binding [75]. By linking to the glycoproteins of the virus the anionic groups of the microalgal polysaccharides induce formation of viral-algal complexes preventing host cell infection [76]. This effect was studied with fucoidan, a sulfated polysaccharide from the marine alga *Cladosiphon okamuranus*, containing glucuronic acid and sulfated fucose residues that exhibited antiviral activity against the dengue virus [77]. Additionally, the in-vitro antiviral activity of sulfated exopolysaccharides produced by the marine microalga *Gyradinium impudicum* was tested against the encephalomyocarditis virus [78] and the influenza A virus [79], showing good antiviral activity in both cases.

Carotenoid-containing extracts of microalgae using hexane, ethanol, and water from *Haematococcus pluvialis* and *Dunaliella salina* were evaluated against *Herpes simplex* virus type 1 at various stages of viral infection [80]. Aqueous extracts of *Haematococcus pluvialis* demonstrated higher antiviral activity, whereas for *Dunaliella salina* the antiviral activity of ethanol extracts was correlated to the presence of short chain fatty acids. As a result, marine microalgae are being further investigated for bioactive compounds [81] and biomedical applications [82].

### 4.7. UV Protection

Light is essential for photosynthesis, but excess light reduces photosynthetic efficiency, therefore microalgae use various mechanisms to dissipate light energy [83]. Increased light intensity can also affect lipid production, but how the biochemical composition of the cell is affected is not well understood yet [84].

Ultraviolet irradiation, in particular, leads to structural damage in DNA, RNA, and metabolic reactions [85], so microalgae defend themselves by boosting carotenoid synthesis, pigment synthesis, and production of antioxidant substances [86]. As a result, microalgal compounds have the potential to provide protection from UV light damage, such ß-carotenes, astaxanthin, astaxanthin, and phycocyanobilins from *Dunaliella salina* and *Haematococcus pluvialis,* and can be used in cosmetics and therapies [87]. Moreover, phycocyanobilins and phycoerythrobilins from *Spirulina* and *Porphyridium* are pigments with an ability to neutralize reactive oxygen species [87].

## 5. Industrial Microalgae Applications

### 5.1. Pharmaceutical Industry

The world market for natural pharmaceutical products is constantly growing and it is expected that algae can meet some of this growing demand. Microalgae biomass is rich in bioactive compounds that are considered difficult to obtain through traditional chemical synthesis and have medicinal, anticancer, antiviral, and antimicrobial potential [10,88]. Primary and secondary metabolites can be used as ingredients for the manufacture of various medicines with the most widely used species being *Arthrospira*, *Chlorella*, *Dunaliella*, and *Haematococcus*.

As outlined in an earlier section, some species of microalgae killed cancer cells during in vitro studies carried out with lung carcinoma, colon cancer, human breast adenocarcinoma, and prostate cancer, suggesting the use of microalgae extracts as possible therapeutics. Moreover, microalgae produce neuroprotective substances that contribute to slowing down or stopping the progression of diseases that affect the nervous system. For instance, *Spirulina platensis* produces neuroprotective substances that exhibited positive results against Alzheimer's and Parkinson's diseases [88].

Several bioactive molecules from microalgae can also be used for antiallergenic therapies because they act as histamine inhibitors with the ability to modulate the immune system, so it is expected that they will be used as functional drugs for the suppression of immune responses in the human body [89] and for development of oral vaccines against influenza A, Zika virus, and HIV viruses [25,26].

### 5.2. Nutraceutical, Cosmetics, and Personal Care Industries

Microalgae can be incorporated in the manufacture of nutraceuticals as they possess certain nutritional and functional characteristics that provide health benefits or reduce the risk of disease. Production of algal nutraceuticals began in Japan in 1960 with the cultivation of *Chlorella* [55]. This species is an important source of nutraceuticals incorporated into daily supplements, functional food, and healthy food. It is known to produce bioactive molecules, such as lutein, astaxanthin, chlorophyll, beta-carotene, fatty acids, and proteins, which are useful to human health [90].

The biological properties of microalgae have also attracted the interest of the cosmetics industry as customers increasingly look for natural products. Microalgae draw attention

for their antioxidant activity that can be used effectively for the treatment of skin disorders, such as aging, pigmentation problems, and UV radiation care by using microalgal substances that stimulate cell renewal and hydration of the skin. Among the personal care products that incorporate microalgal ingredients are exfoliators, facial cleansers, toners with hydration functions, products for skin regeneration, products for sun, oral, hair care, and perfumery. Over 350 cosmetic products derived from algae are currently commercially marketed [25].

Skin protection from UV is one of the most promising industrial applications for microalgae given that they synthesize UV-absorbing compounds for protection of cellular functions. Fucoxanthin, phloroglucinol, and other compounds are included in UV protectors of the keratinous tissue, preventing damage from both long-wave ultraviolet A (UVA) and short-wave ultraviolet B (UVB) radiation and reducing photoaging [4]. They are also used as skin whitening agents because of their ability to inhibit the tyrosinase enzyme and thus reduce melanin pigment production that is responsible for skin and hair coloration.

*Chlorella vulgaris* and *Spirulina Maxima* are used to treat dandruff and stimulate hair growth thanks to their oils [91]. *Spirulina* sp., *Chlorella* sp., and *Arthrospira* sp. are commonly used algae in cosmetics after their bioactive compounds are extracted by conventional methods, including maceration, aqueous extraction, and solvent extraction. Although these microalgae are natural, it is still necessary to carry out studies that validate product safety for microalgal metabolites [26].

### 5.3. Food Industry

Microalgal biomass is a rich source of nutrients, such as carbohydrates, vitamins, proteins, and lipids that are used as additives in food, pasta, dairy products, and baking products approved by the US Food and Drug Administration (FDA). Microalgae represent the main source of omega-3 polyunsaturated fatty acids, mainly decahexaenoic (DHA) and eicosapentaenoic (EPA) acids, for enriching eggs, meat, and milk [92]. The use of microalgae as human food is not new, as it has been part of the daily diet of people for centuries in Africa, Mexico, China, and other Asian countries. The most commercialized algae species are *Chlorella* and *Spirulina* due to their rapid growth and adaptability to various climates. These algae are used for the manufacture of food supplements in the form of tablets or capsules and liquids that are sold as sources of vitamins and antioxidants [93].

Food colorants are extracted typically from *Chlorella* and *Spirulina* species in order to replace synthetic colorants thanks to the nutritional components they contain and the lack of allergic reactions. *Spirulina* sp. are well-known for their high content of pigments and nutrients, such as phycobiloproteins, including phycocyanin, which is a blue pigment that is used in the food industry not only for its striking coloration, but also as a protein source with antioxidant and anti-inflammatory qualities that surpass those of vitamin C [94]. There are also other species of algae that are of interest to the food market due to their properties and characteristics for various consumer applications, such as *Dunaliella*, *Haematococcus*, *Schizochytrium*, *Scenedesmus*, *Aphanizomenon*, *Odontella*, and *Porphyridium* [95].

Recently, new microalgae-based product applications have been developed, such as drinks with *Spirulina* for athletes, pasta with improved nutritional content, oils rich in PUFA (such as EPA and DHA), milk enriched with DHA, yogurt with improved toughness and elasticity, and probiotics [25]. Although microalgae are considered superfoods, their commercialization for the needs of the food industry faces obstacles, such as cultivation scale-up. Large-scale installations for algae production are still capital-intensive and rather complex to operate, while at the same time approval of algae products by regulatory authorities require lengthy testing to ensure consumer safety [96].

### 5.4. Animal Feed and Aquaculture

*Chlorella* was first reported in 1952 as an animal feed ingredient that improved the growth of chicken. Farming animals and pets benefit from the vitamins, minerals, and essential fatty acids of microalgae that improve immune response and fertility, and con-

tribute to healthy skin and external appearance [97]. Moreover, de-fatted marine microalgal biomass, which is the cell residue after lipid extraction for biofuel production, possesses high protein content, a balanced amino acid profile, and high levels of minerals, vitamins, and PUFA that can be used in animal feed. Microalgae nutrients also benefit the aquaculture industry as they are critical for proper development of larvae and adult fish growth and survival. The commonly cultured algal species for aquaculture include *Isochrysis*, *Dunaliella*, *Phaeodactylum*, *Nannochloropsis*, *Tetraselmis*, and *Chlorella*.

### 5.5. Agricultural Fertilizers and Soil Amendments

Microalgae have also found applications in agriculture thanks to their plant biostimulation benefits in the form of soil amendments and biofertilizers that can reduce the use of fossil-derived chemical fertilizers linked to water pollution and eutrophication. Algae biostimulants are capable of promoting the growth and development of various crops under optimal and stressful conditions as they scavenge nitrogen and phosphorus from the ecosystem and make those nutrients available to depleted soils that result from intensive agricultural use [98].

In rice production, a polyculture of *Chlorella vulgaris* and *Scenedesmus dimorphus* was applied to compensate for low nitrogen levels under greenhouse conditions. The results showed that the algae-based biofertilizer resulted in bigger seedlings than chemical fertilizers [99]. When dry powder of *Chlorella vulgaris* was used as fertilizer for the germination of lettuce seeds, it significantly improved the number of plants obtained compared to a control and also led to an increase in pigments in the lettuce [26].

### 5.6. Chemical Products

Microalgae can serve as a renewable source of a wide range of chemical products, including paints, dyes, pigments, bioplastics, biopolymers, and nanoparticles. Microalgae cells produce three main pigment types: chlorophylls, carotenoids, and phycobilins. These compounds can be used to replace synthetic dyes that are derived from fossil sources and may contain impurities, such as lead, that is toxic to humans, and often are allergens and irritants. For example, chlorophyll is used to dye wool and derivatives. Synthetic colorants are recalcitrant and typically not recyclable, whereas microalgal dyes are easily degradable and can help reduce freshwater contamination [100].

Bioplastics, such as polyhydroxybutyrate and polylactic acid, have been produced from microalgae [101]. In the chemical industry the development of bioplastics from microalgae that produce polyhydroxyalkanoates (PHA) is of strong interest. In fact, studies conducted with *Chlorella pyrenoidosa* showed a PHA accumulation of 27%. Synthesis of biopolymers in algae is controlled by enzymatic expression and concentration of nutrients, such as carbon and nitrogen, while low levels of phosphorus stimulate PHA production. However, production costs are still high due to the cost of carbon and nutrient sources, and the energy intensity of the process [16,25].

Another microalgae application is the conversion of algal biomass to hydrochar and biochar, which are carbon-rich materials used in solid fuels and capacitors, for carbon sequestration, and as soil amendments [102,103]. They are both obtained by pyrolyzing algal biomass, but hydrochar is derived from wet biomass at temperatures up to 350 °C, whereas biochar is derived from dry biomass at temperatures up to 600 °C [104]. Actually, defatted algal biomass can be used, instead of whole cells, hence preserving the value of the lipids for higher value applications, such as nutraceuticals, or higher volume applications, such as biofuels. Defatted *Chlorella* sp. were used for production of hydrochar [105] and biochar [106]. Other value-added applications include conversion of algae to biosorbents [107] and activated carbon [108] for water purification and wastewater treatment, which are described in more detail in the next section.

The main cellular biomaterials obtained from a variety of microalgae, their key properties, and their commercial applications are summarized in Table 1.

**Table 1.** Microalgal biomaterials possessing properties of commercial interest.

| Microalgal Biomaterial | Main Compound | Property | Commercial Application | Microalgae Source | References |
|---|---|---|---|---|---|
| Proteins | Phycobiliproteins, glycoproteins | Antitumor, antimicrobial, anti-inflammatory, UV protection | Nutraceuticals, medicines, cosmetics, dyes, food products, animal feed | *Spirulina* sp., *Chlorella vulgaris, Dunaliella salina, Dunaliella* sp., *Haematococcus pluvialis, Porphyridium* sp. | [31,32,55,81,87] |
| Enzymes | Acyl-CoA, diacylglycerol, acyltransferase 2, Δ6-desaturase, L-asparaginase | Antitumor | Production of biofuels, bioremediation, cosmetics, medicines, cleaning products and other chemical products | *Chlamydomonas reinhardtii, Chlorella elipsoidea, Phaeodactylum tricornutum, Chlorella vulgaris, Chlamidomonas* sp. | [30,33,34] |
| Sugars and Polysaccharides | Calcium spirulan, glycosoaminoglycanxylose, glucose, galactose, mannose, cellulose, alginate, agar | Antitumor, anti-inflammatory, antiviral | Nutraceuticals, drugs, cosmetics, food additives, industrial products, biofuel | *Chlorella vulgaris, Porphyridium* sp., *Porphyridium cruentum, Spirulina* sp. | [31,37,55,69,73] |
| Fatty acids and Lipids | Phospholipids, glycolipids, sterols, polyunsaturated fatty acids (PUFAs), decahexaenoic acids (DHA), eicosapentaenoic acid (EPA), chlorellin | Antitumor, antimicrobial, antifungal, anti-inflammatory, antiviral | Nutraceuticals, drugs, food, animal feed, biofuel production | *Chlorella* sp., *Spirulina* sp., *Spirulina maxima, Dunaliella salina* | [11,55,61,68,69,81] |
| Carotenoids and Pigments | Astaxanthin, β-carotene, lutein, fucoxanthin | Antioxidant, antitumor, antifungal, anti-inflammatory, UV protection | Nutraceuticals, medicines, food products, food additives, colorants, manufacture of chemical products | *Haematococcus pluvialis, Dunaliella salina, Scenedesmus* sp. | [3,42,52,68,69,81,87] |
| Polyphenols | Salicylic, transcinnamic, chlorogenic, caffeic acids | Antioxidant, antifungal, anti-inflammatory | Nutraceuticals, food products, drugs and cosmetics | *Spirulina maxima, Phaeodactylum tricornutum, Tetraselmis suecica* | [46,47,68,71] |
| Vitamins | A, C, E, thiamine, riboflavin, niacin, pantothenic acid, pyridoxine, folic acid, cobalamin | Antioxidant, antifungal, antiviral | Nutraceuticals, food, food additives, animal feed ingredient, cosmetics | *Chlamydomonas* sp. *Chlorococcum* sp. *Chlorella vulgaris, Chlorella* sp., *Spirulina* sp. | [48,50–52] |

## 6. Environmental Algae Applications

### 6.1. Carbon Sequestration

Given their high photosynthetic capacity, microalgae are considered a promising way to sequester and recycle atmospheric $CO_2$ through photosynthesis for the production of algal biofuels and bioproducts. The $CO_2$ sequestration efficiency of microalgae is 10–50× higher than terrestrial plants [109]. They are able to remove $CO_2$ from industrial flue gases released during power generation or chemical processing and transform them into organic molecules, like lipids, pigments, proteins, and carbohydrates, as detailed in earlier sections [110]. However, scale-up studies under real-world conditions will have to be conducted to quantify long-term performance and address process challenges, such as the need for flue gas cooling, before microalgae cultivation is integrated with fossil power generation, chemical manufacturing, and biorefining.

### 6.2. Bioremediation

Microalgae are used for the treatment of industrial and mining effluents because of their nutrient removal capability, adsorption capacity, and renewable and sustainable nature compared to chemical treatments. Microalgae are known to use nutrients present in contaminated effluents during their growth and to retain heavy metals, also referred to as potentially toxic elements. The main advantage of microalgae for bioremediation is the

favorable cost-benefit ratio as they can survive in adverse conditions, develop symbiotic interactions with other microorganism within a consortium, adapt to various environments, and be used at an industrial level. On the negative side, their action is limited to aquatic environments and remediation time is longer compared to chemical treatments [111].

Heavy metals released into the environment by anthropogenic processes are persistent and tend to bioaccumulate, thus negatively impacting the environment and human health. Phycoremediation, which is the use of microalgae in environmental pollution treatment, has emerged as an ecofriendly technology for the removal of heavy metals from contaminated effluents. Microalgae cells use biosorption and bioaccumulation as main mechanisms for metal removal. When tested in municipal effluents, removal efficiencies of *Spirulina platensis* for $Cu^{2+}$ and $Ca^{2+}$ ranged from 91 to 98%, *Chlorella minutissima* removed 62% of $Zn^{2+}$, 84% of $Mn^{2+}$, 74% of $Cd^{2+}$, and 84% of $Cu^{2+}$, while *Scenedesmus* sp. and *Chlorella* sp. removed 59 and 56% of $Ca^{2+}$, and 29 and 56% and $Mg^{2+}$, respectively [112].

Abiotic factors play an important role in the remediation of heavy metals by microalgae. The pH is critical for metal biosorption and removal efficiency, since the availability of metal-binding groups is highly dependent on pH. On the other hand, there is no clear relationship between temperature and metal uptake, so the optimum temperature needs to be determined for each strain and metal of interest [113]. Several studies have demonstrated that contact time is important for biosorption by both living and non-living microalgae cells and that the algal content of lipids, carbohydrates, and proteins correlates with metal removal efficiency. For instance, *Chlorella vulgaris* exposed to $1.2 \times 10^{-5}$ M $FeCl_3$ had a lipid content of 56.5% [114]. When *Chlorella vulgaris* was supplemented with $10^{-9}$ M of $Co^{2+}$, $Cu^{2+}$, and $Zn^{2+}$, the total accumulation of carbohydrates was 330, 270, and 310 mg/g dry weight, respectively [115], while the total free amino acids were 0.6, 0.7, and 1.4 mg/g dry weight, respectively [115]. Hence, heavy metals appear to have both stimulatory (micronutrients) and inhibitory effects on the algae cells depending on metal concentration. As a result, microalgae strains have different sensitivities to the same metal, so future research should investigate the development of metal-tolerant algae strains with high specificity and metal removal efficiency. Similarly, there is a need for researching sustainable processes that integrate wastewater treatment and algal biomass production [112].

### 6.3. Wastewater Treatment

Effective remediation processes are required to reduce the concentration of nutrients, such as nitrogen and phosphorus, and organic matter in municipal and industrial wastewaters, hence the use of microalgae has emerged as a promising strategy capable of eliminating pollutants [116]. The removal of nitrogen and phosphorous is very important to prevent eutrophication in receiving water bodies, reduce ammonia toxicity to fish, and prevent interference with free chlorine that is essential for drinking water treatment [117]. The use of microalgae for the removal of nutrients and organic matter from wastewater effluents was recently reviewed [118,119]. When cultivated in wastewater, microalgae achieve high fixation rates of carbon with values up to 1.83 kg $CO_2$/kg of biomass and high biomass productivity that exceeds by 40–50% that of terrestrial crops with reduction efficiencies up to 80–100% of nutrients and 90% of chemical oxygen demand (COD) [119]. The main disadvantages of microalgae are the risk of culture contamination, difficulty of cell harvesting, and limited experience with large-scale processing.

Interestingly, microalgae-bacteria mixed cultures perform well in wastewater treatment due to synergistic action that results in better adaptability, higher nutrient uptake, and higher biomass production under environmental stress conditions [120]. When *Chlorella* sp. and *Scenedesmus* sp. were cultivated with activated sludge in synthetic municipal wastewater, the mixed culture achieved 98% removal of $NH^{4+}$-N and 100% removal of $PO_4^{3-}$-P and COD with a biomass productivity of 0.76 g/L and lipid accumulation of 15.3–16.7% under light and dark conditions [121]. In the same study, when only *Chlorella* sp. and *Scenedesmus* sp. were cultivated in the same effluent, removal efficiencies of the nutrients and lipid accumulation were very similar, but the biomass productivity was

lower at 0.05–0.7 g/L, suggesting that the bacteria improved biomass productivity in the symbiotic system [121]. Based on such findings it appears that wastewater treatment can be integrated with microalgae cultivation in a circular fashion to produce algal biomass and biomaterials, while removing nutrients from wastewater. Future research should focus on conducting cost–benefit analysis of such integrated systems, where wastewater serves as a low-cost abundant growth medium for biomass and algal metabolite production in the context of a biorefinery [119].

### 7. Conclusions

Microalgae are a promising renewable feedstock for the biorefineries of the future to produce biofuels and biomaterials of industrial interest, such as amino acids, proteins, enzymes, sugars, polysaccharides, fatty acids, lipids, carotenoids, pigments, and polyphenols. Numerous microalgal metabolites have shown health and nutritional benefits that have attracted the interest of the pharmaceutical, nutraceutical, cosmetic, and food industries. Other applications of microalgae include stimulation of plant growth and restoration of soil nutrition in agriculture and nutritional value for fish in aquaculture. Moreover, microalgae can advance environment stewardship through bioremediation thanks to their ability to absorb nutrients and retain heavy metals under adverse conditions. Their incorporation in municipal and industrial wastewater treatment can be an effective contributor to preventing eutrophication of natural water resources by reducing nitrogen and phosphorus concentrations. The sustainability potential of microalgae can be further enhanced if they are successfully converted to bioplastics and biopolymers, where there is an urgent global need for natural materials to replace plastics from petroleum. Still, commercialization of algal biorefineries depends on solving technology hurdles associated with both microalgae cultivation and downstream processing in order to achieve the productivity and resiliency needed to make the technology cost-competitive. In that context, we view microalgal biomaterials and environmental applications as a crucial enabler of the commercial success of microalgal biofuels.

**Author Contributions:** All authors contributed to conceptualizing, writing, and editing the manuscript. All authors have read and agreed to the published version of the manuscript.

**Funding:** Partial financial support to G.P.P. was provided by the Florida Department of Agriculture and Consumer Services through Grant Agreement 26100. L.O.-E. would like to thank the Dean of Research and the Dean of the College of Sciences and Engineering at Universidad San Francisco de Quito (USFQ) for supporting her research activities. Financial assistance was granted to V.O.-H. by USFQ through Collaboration Grant 16890.

**Institutional Review Board Statement:** Not Applicable.

**Informed Consent Statement:** Not Applicable.

**Data Availability Statement:** Not Applicable.

**Conflicts of Interest:** The authors declare no conflict of interest.

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
