# Peer review of "Prospects of Microalgae for Biomaterial Production and Environmental Applications at Biorefineries"

_sustainability, doi:10.3390/su13063063_

Round 1

Reviewer 1 Report

Manuscript ID: sustainability-1115001

Type of manuscript: Review

Title: Prospects of Microalgae for Biomaterial Production and Environmental

Applications at Biorefineries

Authors: Lourdes Orejuela-Escobar *, Arleth Gualle, Valeria Ochoa-Herrera,

George Philippidis

Submitted to section: Energy Sustainability,

Manuscript deals with microalgae as a source of chemical compounds, potential use of those compounds in the industry like food, pharmaceutical and chemical industry, and use of microalgae in wastewater treatment, bioremediation, and biofuel production.

The manuscript's topic is within the journal's scope and presents scientific achievements that is consistent with Sustainability. However, some revisions are needed as it is now.

Special Comments:

Page 1, lies 32-35

„They have certain requirements for growth, such as light, nutrients, and appropriate pH, and are capable of obtaining energy from photosynthesis and sugars, when growing under photoauto trophic and heterotrophic or mixotrophic conditions, respectively [2].“

Some microalgae can obtain energy from sugars as states in text. However, microalgae can use other types of carbon source like acetate, glycerol, malate and citrate, in the presence or absence of a light supply. I would suggest replacing “sugars” with “organic carbon” or something similar. Please revise.

Page 2, lines 53-56

„The cell wall is rigidly built with phospholipids and polysaccharides, like cellulose and hemicelluloses,  that necessitate physical and chemical processing, such as shearing, cavitation, pulsed electric field, hydrolysis, enzymatic digestion, extraction with supercritical fluids, use of eutectic solvents, and high-pressure homogenization [4].“

The cell wall of microalgae is generally is composed of glycoproteins, cellulosic material, carbonate or silica. Plasma membrane is rich in phospholipids and proteins. Please revise the text.

Page 3, figure 1

“2nd generation biorefinery, lignocellulosic biomass (forest, agricultural, municipal, etc. residues)”

Composition of municipal waste varies widely depending on the pattern of consumption, lifestyles, the rate of urbanisation, etc. It’s mainly composed of organic matter such as lipids, carbohydrates, proteins. Please revise the text.

Page 4, figure 3.

The authors classified high-pressure homogenisation under heat treatment. Destruction of cell wall by high-pressure homogenisation is achieved

by mechanical forces. As the cells are forced at high pressure through the orifice, they are subjected to a combination of cavitation and liquid shear. During a process, the temperature is usually controlled. Please revise the text.

Figure 3. instead of “Microwave oven” authors should consider using “Microwave –assisted extractions”.

Figure 3. Lipids are used as feedstock for synthesis of biodiesel. However, additional income could be obtained in biorefinery if high-value PUFAs are separated before biodiesel production. Authors should consider adding this step to the scheme of microalgae biomass processing.

Page 5,lines 150-151

“Acids generate a rapid increase in temperature that causes degradation of cell wall polypeptide chains and depolymerization of starch and other polysaccharides.”

Acid treatment is conduct at higher and lower temperatures, using relatively low acid concentrations. Higher temperatures usually lead to a higher degree of cell disruption than the same treatments at lower temperatures. Treatment with acids leads to poration of the cell wall and release the cell content. Please revise the text.

Page 5, lines 171-175

Numerous enzymes, which are proteinaceous molecules that catalyze metabolic reactions, are synthesized in microalgae [27]. They include cellulases, lipases, amylases, galactosidases, proteases, phytases, and laccases. The presence of cellulases, hemicellulases,  and pectinases creates capacity to degrade polysaccharides. Amylases, such as β-amylase, glucoamylase, isoamylase, and glucosidases promote the hydrolysis of starch. The enzyme galactosidase hydrolyzes galactose residues linked to the α-1,6 bond found in oligosaccharides and galactomannans [30].

All living creatures, including microalgae, synthesise different classes of enzymes (including proteinase, lipase, carbohydrate) involved in cell metabolism. Therefore, I suggest adding the list of specific industrially interesting enzymes from microalgae (e.g. enzymes with interesting substrate specificity, thermostable enzymes, enzymes stable in organic solvents or at extreme pH).

Page 5, Lines 186-195

Monosaccharide composition of the most abundant carbohydrates in microalga is mentioned in the text. Since the focus of the review is an application of bioactive compounds, it would be interesting to include types of polysaccharides isolated from microalgae, composition and specific application (e.g. in the form of a table). Please consider including these data.

Page 5, lines 196-199

The lipids are located in lipid droplet, a globular organelle (enclosing TAG as a major constituent). Lipid droplets are covered by a membrane consisting of phospholipids and specific proteins. A homologous organelle is found in plants, algae, oleaginous yeasts and fungi. Please revise the text.

Page 8, lines 306-308

„Recently, antifungal activity was reported in extracts of Nannochloropsis oculata and Candida albicans, where the bioactive compounds were terpenoids, carotenoids, polyphenols, and fatty acids [64]“.

In this paper, the antimicrobial activity of extracts of Nannochloropsis oculata against yeast Candida albicans was measured. Please revise the sentence.

Author Response

Please notice that changes were highlighted (in yellow) in the revised manuscript.

  1. Page 1, lines 32-35

„They have certain requirements for growth, such as light, nutrients, and appropriate pH, and are capable of obtaining energy from photosynthesis and sugars, when growing under photoautotrophic and heterotrophic or mixotrophic conditions, respectively [2].“

Some microalgae can obtain energy from sugars as states in text. However, microalgae can use other types of carbon source like acetate, glycerol, malate and citrate, in the presence or absence of a light supply. I would suggest replacing “sugars” with “organic carbon” or something similar. Please revise.

As suggested by the reviewer, we used the term “organic carbon”.

  1. Page 2, lines 53-56

„The cell wall is rigidly built with phospholipids and polysaccharides, like cellulose and hemicelluloses,  that necessitate physical and chemical processing, such as shearing, cavitation, pulsed electric field, hydrolysis, enzymatic digestion, extraction with supercritical fluids, use of eutectic solvents, and high-pressure homogenization [4].“

The cell wall of microalgae is generally is composed of glycoproteins, cellulosic material, carbonate or silica. Plasma membrane is rich in phospholipids and proteins. Please revise the text.

As suggested by the reviewer, the text was revised.

  1. Page 3, figure 1

“2nd generation biorefinery, lignocellulosic biomass (forest, agricultural, municipal, etc. residues)”

Composition of municipal waste varies widely depending on the pattern of consumption, lifestyles, the rate of urbanisation, etc. It’s mainly composed of organic matter such as lipids, carbohydrates, proteins. Please revise the text.

As suggested by the reviewer, the text was revised.

  1. Page 4, figure 3.

The authors classified high-pressure homogenisation under heat treatment. Destruction of cell wall by high-pressure homogenisation is achieved by mechanical forces. As the cells are forced at high pressure through the orifice, they are subjected to a combination of cavitation and liquid shear. During a process, the temperature is usually controlled. Please revise the text.

Figure 3. instead of “Microwave oven” authors should consider using “Microwave –assisted extractions”.

Figure 3. Lipids are used as feedstock for synthesis of biodiesel. However, additional income could be obtained in biorefinery if high-value PUFAs are separated before biodiesel production. Authors should consider adding this step to the scheme of microalgae biomass processing.

We made changes to Figure 3 as suggested by the reviewer.

  1. Page 5,lines 150-151

“Acids generate a rapid increase in temperature that causes degradation of cell wall polypeptide chains and depolymerization of starch and other polysaccharides.”

Acid treatment is conduct at higher and lower temperatures, using relatively low acid concentrations. Higher temperatures usually lead to a higher degree of cell disruption than the same treatments at lower temperatures. Treatment with acids leads to poration of the cell wall and release the cell content. Please revise the text.

As suggested by the reviewer, the text was revised.

  1. Page 5, lines 171-175

Numerous enzymes, which are proteinaceous molecules that catalyze metabolic reactions, are synthesized in microalgae [27]. They include cellulases, lipases, amylases, galactosidases, proteases, phytases, and laccases. The presence of cellulases, hemicellulases, and pectinases creates capacity to degrade polysaccharides. Amylases, such as β-amylase, glucoamylase, isoamylase, and glucosidases promote the hydrolysis of starch. The enzyme galactosidase hydrolyzes galactose residues linked to the α-1,6 bond found in oligosaccharides and galactomannans [30].

All living creatures, including microalgae, synthesise different classes of enzymes (including proteinase, lipase, carbohydrate) involved in cell metabolism. Therefore, I suggest adding the list of specific industrially interesting enzymes from microalgae (e.g. enzymes with interesting substrate specificity, thermostable enzymes, enzymes stable in organic solvents or at extreme pH).

As suggested by the reviewer, the text was revised to add names of enzymes.

  1. Page 5, Lines 186-195

Monosaccharide composition of the most abundant carbohydrates in microalga is mentioned in the text. Since the focus of the review is an application of bioactive compounds, it would be interesting to include types of polysaccharides isolated from microalgae, composition and specific application (e.g. in the form of a table). Please consider including these data.

As suggested by the reviewer, types of polysaccharides were added and a table was created (Table 1) to summarize pertinent information.

  1. Page 5, lines 196-199

The lipids are located in lipid droplet, a globular organelle (enclosing TAG as a major constituent). Lipid droplets are covered by a membrane consisting of phospholipids and specific proteins. A homologous organelle is found in plants, algae, oleaginous yeasts and fungi. Please revise the text.

As suggested by the reviewer, the text was revised.

  1. Page 8, lines 306-308

„Recently, antifungal activity was reported in extracts of Nannochloropsis oculata and Candida albicans, where the bioactive compounds were terpenoids, carotenoids, polyphenols, and fatty acids [64]“.

In this paper, the antimicrobial activity of extracts of Nannochloropsis oculata against yeast Candida albicans was measured. Please revise the sentence.

As suggested by the reviewer, the text was revised.

Reviewer 2 Report

Dear Authors, the paper is interesting and has a good potential.

The topic is very trendy and a lot of publications have been published in the last two, three years, the reference list surely adheres to that.

The text is comprehensive and well written. I would suggest to strenghten the abstract by defining a nische for the review.

The resolution of figures 1 and 2 needs to be improved. As it is not an original finding, also the figures should be cited.

A consequent approach and discussion is carried throughout the paper, somethimes the Authors fail to review the topic critically, however overall the conclusiond are given accurately.

Author Response

Please notice that changes were highlighted (in yellow) in the revised manuscript.

  1. The text is comprehensive and well written. I would suggest to strengthen the abstract by defining a niche for the review.

As suggested by the reviewer, the abstract was revised to explicitly state that the review article focuses on the use of microalgae for biomaterial production and environmental applications.

  1. The resolution of figures 1 and 2 needs to be improved. As it is not an original finding, also the figures should be cited.

The resolution of the figures in the submitted material appears to be reasonable. If not, we will be glad to seek ways to improve it.

  1. A consequent approach and discussion is carried throughout the paper, sometimes the Authors fail to review the topic critically, however overall the conclusions are given accurately.

We appreciate the reviewer’s comment. The breadth of the topics covered in the manuscript may make some sections look more or less critically reviewed than other sections.

Reviewer 3 Report

The paper entitled“ Prospects of Microalgae for Biomaterial Production and Environmental Applications at Biorefineries” is a review paper dealing with a recognition of the potential products coming from microalgae as well as potential applications of microalgae for facing environmental issues (i.e. CO2 sequestration, phycoremediation and nutrients removal from wastewater. The paper is well written and properly structured to be easily consulted. Such review can be very useful for readers because they can find in it the most relevant aspects concerning the variety of microalgae based products and processes. To improve the paper and give it more strength I have some suggestion as follow:

  • Provide the paper with a table where the main aspects examined in this review are summarized. For example, in first column the most relevant microalgae based products and processes could be reported and in the following ones their characteristics, their applications, the species that are more interesting for being used to produce that specific product and/or performing that specific process, etc… and the final column the reference papers;
  • In the introduction, in particular in the sentence at lines 59-63, one of the critical aspects for microalgae cultivation, i.e. their harvesting, is not mentioned. For more details have a look at these papers: 10.1016/j.jenvman.2018.04.010, 10.1016/j.jenvman.2019.109957 and 10.3390/su12219083. Anyway, many other papers about this topic can be found in the scientific literature;
  • The order of subsections could be modified as follow: i) subsection 3.2 could be moved to the end of section 3; ii) subsection 4.3, 4.6 and 4.4, could become 4.1, 4.2 and 4.3 respectively, then 4.1, 4.2, 4.5, could become 4.4, 4.5, 4.6, respectively. Whereas 4.7 will not be moved;
  • Recent scientific literature suggests the use of the term potentially toxic elements (PTEs) in place of heavy metals. For more details, have a look at this paper: 10.3390/ijerph16224446. Therefore replace in the entire manuscript “heavy metals” with “potential toxic elements”. For example, at line 23, 508, 514, 517, 523, 534 and 579;

Author Response

Please notice that changes were highlighted (in yellow) in the revised manuscript.

  1. Provide the paper with a table where the main aspects examined in this review are summarized. For example, in first column the most relevant microalgae based products and processes could be reported and in the following ones their characteristics, their applications, the species that are more interesting for being used to produce that specific product and/or performing that specific process, etc… and the final column the reference papers.

As suggested by the reviewer, a table summarizing relevant information was created and added to the revised manuscript (Table 1).

  1. In the introduction, in particular in the sentence at lines 59-63, one of the critical aspects for microalgae cultivation, i.e. their harvesting, is not mentioned. For more details have a look at these papers: 10.1016/j.jenvman.2018.04.010, 10.1016/j.jenvman.2019.109957 and 10.3390/su12219083. Anyway, many other papers about this topic can be found in the scientific literature.

As suggested by the reviewer, key information and references on cell harvesting were added to the introduction (after the narrative on cultivation and before cell disruption).

  1. The order of subsections could be modified as follow: i) subsection 3.2 could be moved to the end of section 3; ii) subsection 4.3, 4.6 and 4.4, could become 4.1, 4.2 and 4.3 respectively, then 4.1, 4.2, 4.5, could become 4.4, 4.5, 4.6, respectively. Whereas 4.7 will not be moved.

Although we agree with the reviewer that subsections can be rearranged, the flow in the mentioned sections of the manuscript appears to be reasonable.

  1. Recent scientific literature suggests the use of the term potentially toxic elements (PTEs) in place of heavy metals. For more details, have a look at this paper: 10.3390/ijerph16224446. Therefore replace in the entire manuscript “heavy metals” with “potential toxic elements”. For example, at line 23, 508, 514, 517, 523, 534 and 579.

We use the term “heavy metals” as it is widely established in the literature from which we drew information for this manuscript. Nevertheless, in accordance with the reviewer’s comment, we added the term “potentially toxic elements” in section 6.2 (Bioremediation) to account for this recent trend.

Round 2

Reviewer 1 Report

The manuscript has been improved, but still, minor revision is needed.

Page 5, Figure 3

From Fig. 3 seems that biodiesel is produced only from PUFA. Usually, low-value fatty acids (SFAs, MUFAs) are used as feedstock for the production of biodiesel and not PUFAs. Some of PUFAs are biologically active compounds with a high market price. Thus, it would seem more reasonable to use only low-value fatty acid for the production of biodiesel. Please revise Figure 3 (PUFA).

Page 6, lines 191-194

“Microalgae contain enzymes with industrial utility for production of lipids and biodiesel, such as acyl-CoA, diacylglycerol, acyltransferase2, and ∆6-desaturase, which can be obtained from a variety of species, such as Chlamydomonas reinhardtii, Chlorella elipsoidea, and Phaeodactylum tricornutum [30].”

Microalgae contain enzymes need for a synthesis of fatty acids (de novo assembly of acetate into saturated fatty acids) and enzymes for further modification through desaturation and elongation. Neutral lipids are synthesised by esterification of glycerol with fatty acids (lipases). However, biodiesel is produced from triacylglycerols by alkali-catalysed transesterification in the presence of methanol/ethanol. Please revise the above sentence.

Author Response

We appreciate the reviewer's diligent review and have made corrections to both points, as noted in the re-revised manuscript:

(1) Figure 3 on p. 5 was edited to reflect the recovery of high value compounds, such as PUFA, for use in high-end applications, whereas the lower value lipids can be converted to biodiesel via transesterification. At the same time we also corrected some typos in that figure.

(2) We revised the sentence in the "Enzymes" paragraph on p. 6 in accordance with the reviewer's guidance about lipid biosynthesis and production of biodiesel (after lipids are extracted).